# Improved Acoustic Thermometry for Long-Distance Temperature Measurements

**DOI:** 10.3390/s23031638

**Published:** 2023-02-02

**Authors:** Marco Pisani, Milena Astrua, Massimo Zucco

**Affiliations:** Istituto Nazionale di Ricerca Metrologica, INRIM, 10135 Torino, Italy

**Keywords:** long-distance measurements, interferometry, thermometry, speed of sound, aerospace manufacturing, laser trackers, air temperature gradient

## Abstract

Accurate measurements of long distances (in the order of tens of meters or more) are necessary in manufacturing processes of large structures, as, for example, in the aerospace industry. In the most demanding applications, the goal is to achieve a relative accuracy of 10^−7^ in the measurement of distances (e.g., 1 µm over 10 m). This goal can be obtained with laser interferometers whose accuracy is based on knowledge of the speed of light, which, in turn, depends on the temperature of air. A thermometer based on the measurement of the speed of sound in air has been realized at INRIM. Its purpose is the measurement of the air temperature along the measurement path of the interferometer with an accuracy of 0.1 °C at distances up to 11 m. The paper describes the principle and the experimental setup of the acoustic thermometer and demonstrates its performance by comparison with calibrated reference platinum resistance thermometers. Furthermore, we demonstrate the potentiality of the method to measure the vertical temperature gradient, which is the main error source in triangulation measurements when using laser trackers.

## 1. Introduction

Long-distance measurements rely on laser interferometry, or time-of-flight (ToF) measurements such as those realized by electronic distance meters (EDM) [1]. Both methods require an accurate estimate of the speed of light in air, i.e., the refractive index of air, which is affected by several environmental parameters such as (in decreasing order of sensitivity): temperature *T*, relative humidity *RH*, pressure *P,* and *x*CO_2_ concentration. Based on the Edlen formula for the refractive index of air [2,3,4], or on the equations proposed by Ciddor or by Bonsch and Potulski [5,6], if the target relative uncertainty of the distance measurement is set to Δ*L*/*L* = 10^−7^, the quantities defining the thermodynamic state of air need to be known with uncertainties Δ*T* ≈ 0.1 °C, Δ*RH* ≈ 12%, Δ*P* ≈ 40 Pa, and Δ*x*CO_2_ ≈ 1000 ppm. At ordinary ambient conditions, the value and the rates of change of *P*, *RH,* and *x*CO_2_ are all subject to rapid diffusive processes towards local equilibrium, so that single localized measurements of these parameters are representative of a relatively large surrounding environment. Otherwise, temperature can change rapidly and significantly in space and time, due to heat conduction processes, especially outdoors, and a measurement of temperature at a single point in space is scarcely representative of a large volume. 

When measurements of length are performed in the laboratory environment, it is rather easy to obtain an estimate of the average temperature along the measurement axis with uncertainty less than 0.1 °C based on the readings of a few calibrated thermometers. However, when dealing with long-distance measurements, such as those needed in large assembly plants or even outdoors, to achieve a low uncertainty determination of temperature may require a large number of thermometers to be aligned along the optical path. Furthermore, classical thermometers are sensitive to radiative sources, air flux, and self-heating, leading to errors larger than 0.1 °C [7,8]. For these reasons, long-distance measurements based on the speed of light are typically limited to achieving relative accuracy not better than 10^−6^ [9].

A promising method to overcome this limit is acoustic thermometry (AT), which exploits the strong influence of air temperature on the speed of sound. Different methods and instruments have been developed for this purpose. One of the first was presented in [10], using pulses of 50 kHz emitted by acoustic piezo transducers. The temperature range in this work is limited to 19.5–21.0 °C and the reported uncertainty is 25 mK in a distance range limited to 12 m. 

Another recent work describes a device based on the integration of an acoustic thermometer and a tunable diode laser absorption spectrometer able to carry out fast measurements of relative humidity and temperature with a repeatability of 0.1 °C and difference with respect to conventional sensors less than 1 °C [11].

In our previous work [12], we presented a method based on the use of a continuous sound wave, and demonstrated its effectiveness and sensitivity with several experiments at different distances using acoustic waves at different frequencies. 

The present work was carried out in the framework of a European research project [13] aimed at improving the accuracy in large volume manufacturing, where one of the main targets is “to develop and demonstrate techniques for in situ high accuracy (~10^−7^) air refractive index determination”. With this aim in mind, we realized the acoustic thermometer described in this paper. 

In the following sections, we describe experiments suited to test and assess the overall accuracy of the method, i.e., the capability to assess the average temperature in a given air volume without the need of reference thermometers. The results demonstrate that, for indoor distance measurements of the order of 10 m, the agreement goal of 0.1 °C (needed to reach the 10^−7^ air refractive index determination) with respect to calibrated reference platinum resistance thermometers is successfully achieved. Finally, we used the method to measure the vertical temperature gradient in a given air volume with an uncertainty better than 0.1 °C/m.

For a clearer evaluation of the following, it must kept in mind that a 0.1 °C temperature change corresponds to about 0.06 m/s change in the speed of sound around 20 °C that is about 1.7 × 10^−4^ of the speed of sound. This is the target uncertainty we need to achieve.

## 2. Working Principle

The measurement principle is based on the determination of the phase delay *φ* between the generation and the detection of an acoustic wave at a given frequency *f* over a given distance *d*. The measurement directly provides the speed of sound *u* at temperature *T* according to: *u (T)* = *d* · *f*/*φ*(1)

In Figure 1, the principle of the experimental setup is sketched, which basically comprises a loudspeaker emitting a continuous wave, a microphone, and a phasemeter that measures the phase difference between the electrical signal fed to the loudspeaker and the one received at the microphone location. Due to its cyclic nature, the phase signal has an ambiguity, which is derived from the lack of knowledge of the integer number of phase cycles (i.e., the number of acoustic wavelengths) due to the delay between loudspeaker and microphone. Furthermore, the distance *d* is not easily estimated by the knowledge of the physical position of the loudspeaker and microphone, because the dimensions of both devices are larger than the acoustic wavelength, with the consequence that the starting point and the ending point of the acoustic wave are merely “virtual” positions. From now on, we call the “acoustic distance, *d_a_*” the distance *d* used in (1). In contrast, we call “mechanical distance, *d_m_*” the distance between the loudspeaker and the microphone measured with respect to some “fiducial part” of the physical devices, e.g., the distance between the flat surface of the horn and the stand of the microphone.

In order to eliminate both ambiguities (i.e., the integer number of wavelength and the value of the acoustic distance), we developed the following method. By sweeping the frequency in a given interval, in the setup described above, the phase changes continuously proportionally to the distance *d_a_*. It is easy to see that the relationship between the speed of sound, frequency, phase, and distance is:*d_a_* = *u* · *δφ*/*δf*(2)
thus, assuming the linearity of the function *φ(f)*, a measurement of the slope *δφ/δf* allows us to determine the value of *d_a_* to be used in Equation (1). One could observe that the solution of Equation (2) requires the a priori knowledge of *u*, so that Equations (1) and (2) are recursive, preventing a solution from being found. To overcome this problem, once the hardware of the experimental setup was defined, we estimated the value of *d_a_* with the sweep method, in a short distance condition (e.g., 1 m) where the environmental parameters (*T*, *RH*, and *P*) could be accurately measured and the value of *u* can be estimated with a low uncertainty (better than 0.05 m/s). Now, once we established the value of *d_a_*, we measured the distance between the fiducial points *d_m_* and assessed the value *d_a_* − *d_m_* that we assume to be a constant of the setup. From then on, we used these fiducial points to estimate the distance *d_m_*. For this purpose, in the experiments discussed below we used an EDM (Bosh GLM 250) with a resolution of 0.1 mm up to a distance of 10 m. The EDM was calibrated at INRIM with the long range interferometer, LORI [14]. 

## 3. Choice of the Operating Acoustic Frequency

In principle, for the implementation of the acoustic thermometer described in the previous section, an apparatus employing acoustic waves at arbitrary frequency can be used. In fact, the theoretical dependence of speed of sound in air from frequency, as a consequence of accounting for relaxation effects in the acoustic model [15], is typically so small that it can safely be neglected in the present uncertainty context. On the other hand, the choice of one particular working frequency has important implications on several experimental features and specifications such as the band-pass and directionality of the loudspeaker and microphone, the attenuation of the acoustic signal, the resolution achievable by the measurement, and the impact on human hearing. Firstly, we must consider that the shorter the wavelength, the higher the resolution, so higher frequencies are preferable; on the other hand, the attenuation increases dramatically with frequency so, for long distances, low frequencies are better (see Figure 2). Finally, we have to consider that the use of high frequencies in the audible range (1–10 kHz), because of the high sensitivity of human hearing in this range, might be dangerous and annoying for people working in the surroundings. All this considered, in order to increase the measurement resolution, we decided to work in the near ultrasound region, considering that the large attenuation can be partially mitigated by the high directionality that can be obtained in this frequency range by using directional loudspeakers and microphones. Thus, in the present work we operated at frequencies close to 20 kHz, though we remark that analogous results can be obtained with lower frequencies that would better suit the application of the acoustic thermometer to the measurement on larger (50 m to 100 m) distances.

## 4. Experimental Setup

The experiment was set up in the large semi-anechoic room available at INRIM. The room is 12 × 9 m wide and 5 m height. The loudspeaker and the microphone were mounted on tripods at a height of 1.3 m. In the course of preliminary tests, we tested several types of loudspeakers (dynamic, focused, horn, piezo), and several types of condenser microphones (simple capsule, focused, super-cardioid) with very similar results. For large distances, a super-cardioid microphone was preferably used because of its higher directionality. The measurement principle was tested at various distances varying from 1 m to the maximum distance available along the diagonal of the chamber that is 11 m. Four platinum resistance thermometers with nominal resistance of 100 Ω at 0.01 °C (PT100) were used to sample the air temperature along the measurement axis. These thermometers were previously calibrated by comparison to a primary standard thermometer. The calibration was performed in a thermostatic bath at ambient temperature with a resulting expanded (k = 2) uncertainty of 0.05 °C. The thermometers were acquired by a Fluke 1586A Super-DAQ. In Figure 3 and Figure 4, a sketch and a picture of the experimental setup are presented, respectively. 

In addition to the already mentioned loudspeaker and microphone, the instrumentation included a frequency synthesizer to generate the 20 kHz tone and the frequency sweep, a power amplifier to feed the loudspeaker, a 16 bit two channel analog-to-digital converter to acquire the signals coming from the frequency synthesizer and the microphone, and a LabView^®^-based program to elaborate the signals. 

## 5. Results and Discussion

In this paragraph, we illustrate and discuss some representative experimental results obtained with the acoustic thermometer (AT). In both experiments we used a piezo horn loudspeaker and a super-cardioid microphone with built-in preamplifier. The working frequency was set to 20 kHz. The acoustic transducers were placed on tripods at fixed distances with the PT100 thermometers placed on separate tripods at the same height. The ambient pressure and the relative humidity were recorded simultaneously using a Druck DPI 142 barometer with an accuracy of 0.01% and a Testo 440 digital probe having an uncertainty of ±3%, respectively. All measurement recordings lasted several hours and the sampling time of all instruments was set to be 10 s in the first experiment and 1 s in the second one. In both experiments, the distance *d_m_* was measured with the EDM at the beginning and at the end of the recording. The maximum difference between the two measurements was found in the order of 1 mm over several meters. The mean value of the two was taken as the nominal distance value for the whole record. In Figure 5, a picture of the setup is shown. 

In the first experiment, the distance was about 8.2 m. We measured the speed of sound with the AT and calculated the speed of sound from the recorded environmental (*P, T, RH*) parameters according to the prediction of the Cramer equation [16]. Figure 6a,b plot the results of the acoustic measurements, the predictions of Cramer equation, and the difference between the corresponding speed of sound values. For the recording plotted in Figure 6a, the heating resistors of the conditioning system of the anechoic chamber were repeatedly switched on and off manually. During the over nine hours of measurement, the temperature varied in the range between 15 °C and 20 °C, the *RH* range was (17–33) %, and the pressure varied between 99.5 kPa and 99.8 kPa. The resulting agreement between measured and predicted speed of sound is within ±0.1 m/s when the temperature is changes more slowly and increases up to ±0.3 m/s when the temperature changes rapidly. It is easy to attribute this difference to the thermal inertia of the PT100, which takes time to reach the thermal equilibrium with the surrounding air compared to the nearly instantaneous response of the AT. This is confirmed by the experiment in Figure 6b where the conditioning system was left switched off. The measurement took about three days and a progressive cooling of the ambient temperature with superposed natural day/night periodic variations were observable. In this case, the temperature range was (12–18) °C, the *RH* range was (25–47)%, and the pressure range was (99.6–100.2) kPa. For this measurement run, the difference between the measured and predicted speed of sound values is found to be less than 0.06 m/s, corresponding to the target temperature uncertainty of 0.1 °C. 

In a second experimental run, we increased the measuring distance up to about 11 m (roughly the maximum distance allowable in the chamber) to test the overall performance of the AT compared with traditional platinum resistance thermometry. The experiment is conceptually identical to that discussed above except for the increased distance, which raises the noise of the AT due to the turbulence of the air. The result is shown in Figure 7, where the temperature estimated from the speed of sound, *RH*, and pressure measurements is compared to the mean temperature measured by the PT100 readings with the difference between the two thermometry methods reported in the same plot. For this comparison run, the anechoic chamber was initially cooled down to a minimum by exposing it to the winter outdoor conditions; subsequently, the conditioning system was switched on with the heating resistors set to the maximum power. Again, we observe a larger difference between the acoustic and the platinum thermometry when the temperature varies rapidly. On the other hand, when the temperature change is slow enough for the platinum thermometers, the agreement between the two thermometric techniques is within the target of 0.1 °C. 

The results demonstrate that the AT manages to achieve the targeted performance in laboratory conditions in the case of limited air turbulence. We expect that in an industrial environment, with higher daily temperature excursion and more significant air turbulence conditions, the measurement noise will increase, although without affecting the average value of the estimated air temperature. On the contrary, the presence of an air flow, e.g., from air conditioning plants, would introduce a non-negligible error in the speed of sound measurement, affecting the estimated temperature value. First of all, let us consider an air flow with velocity *v_f_* along a direction perpendicular to the measurement axis of the AT. Following the discussion reported in [17], the measured speed of sound *v_m_* differs from the actual speed of sound *v* according to the following model:(3)$$v_{m}=\sqrt{\;v^{2}-v_{f}^{2}}$$

As an example, a transverse air flow with *v_f_* = 2 m/s in our setup would cause an error of the measured speed of sound of the order of 0.006 m/s, resulting in a temperature error of 0.01 °C. Hence, the transverse air flow is not a limiting factor for the accuracy of the AT, at least for flow with a velocity less or equal to 2 m/s.

In case the air flows along the measurement axis of the AT, the speed of the acoustic wave would increase or decrease depending on whether the flow is in the same direction or opposite to the propagation of the acoustic wave. Therefore, this effect results in a limiting factor for the accuracy of the AT. This issue could be overcome by adding a second system for the measurement of the speed of sound arranged in the opposite direction. Therefore, the error caused by a parallel air flow would be removed by the average value of the speed of sound measured by the two systems. 

## 6. Measurement of Vertical Temperature Gradient

In the following experiment, we demonstrated the potentiality of the acoustic thermometer in measuring the average vertical temperature gradient over long distances. The experiment was carried out in the metrological gallery along a distance of about 26 m. Two identical systems, made as described above, worked in parallel at two different elevations from the floor. In order to avoid interference between the systems, the two thermometers worked at different frequencies, namely, 19,370 Hz and 18,020 Hz. A set of 20 thermometers was distributed in two series along the path at two different elevations from the floor in order to measure the average vertical gradient. A sketch of the experimental setup for the measurement of the vertical temperature gradient is shown in Figure 8 and a picture of the experiment is shown in Figure 9. 

In Figure 10, the temperatures measured with the two thermometers (acoustic and classic) at two elevations are shown. The conditioning plant of the laboratory was switched off in order to allow a small vertical gradient to be generated. In Figure 11, the vertical gradients measured with the two methods are compared. An average gradient of about 0.6 °C/m is observed with both techniques and the agreement between the methods is better than 0.1 °C. As a comment to this result, we wish to point out that the agreement could be even better, since the uncertainty of the thermistors is 0.1 °C and the volume measured by the thermometers is not exactly coincident with the volume travelled by the acoustic wave. The same experiment was carried out successfully on a distance of 60 m, although in this case a comparison with classic thermometers was not performed.

## 7. Conclusions

We realized a method capable of measuring the speed of sound in air at a scale of 10 m or more. The method allows for estimating the average temperature of air along the acoustic path. The effectiveness of the method is demonstrated by comparing the acoustic temperature measurement with the same temperature measured by classical calibrated platinum thermometers. The agreement between the two types of thermometry is of the order of 0.1 °C over a distance of 11 m in a temperature range between 12 °C and 20 °C. These results indicate that the refractive index of air and, hence, interferometric distance measurements, can be performed with a relative accuracy of 10^−7^. Furthermore, the effectiveness of the method in measuring vertical temperature gradients in a large volume by using two acoustic thermometers working in parallel is demonstrated. The latter result is of particular importance in the case of distance measurements based on triangulation techniques, such as the ones made with theodolites or with laser trackers. Finally, the capability of working with two acoustic thermometers at the same time allows for the operation of two devices in opposite directions in order to reduce the error induced by air flux.

## Figures and Tables

**Figure 1 sensors-23-01638-f001:**
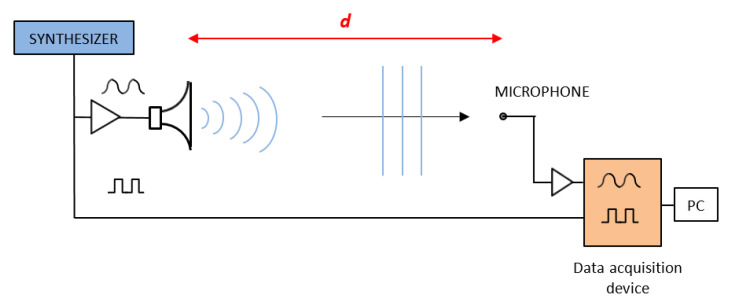
Principle of the experiment.

**Figure 2 sensors-23-01638-f002:**
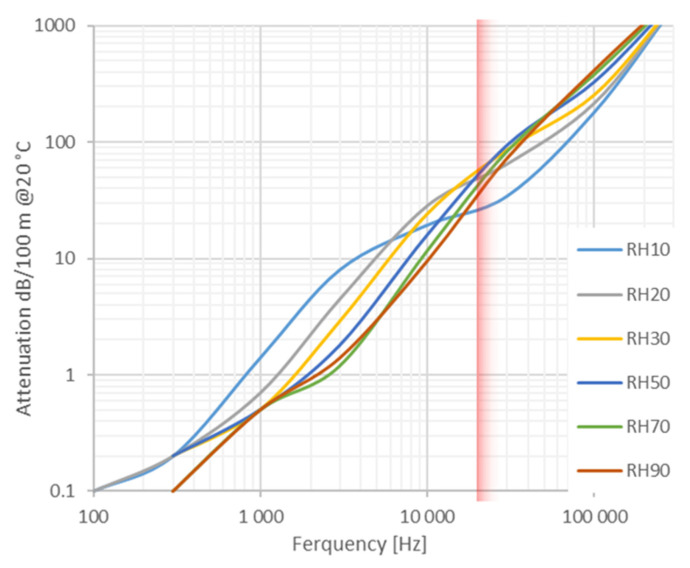
Attenuation of the acoustic wave vs. frequency and for different relative humidity (*RH* from 10% to 90 %) calculated from the ISO 9613-1:1993. It is evident that high frequencies have the disadvantage of a strong attenuation also depending on relative humidity. The red area indicates the operative frequencies chosen for the instrument.

**Figure 3 sensors-23-01638-f003:**
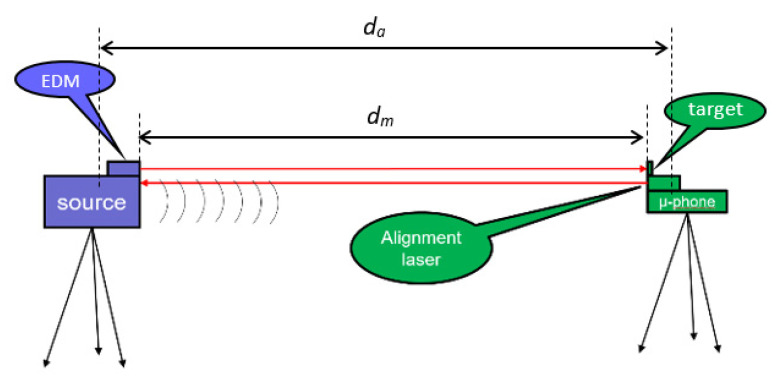
A simplified sketch of the acoustic thermometer. Two tripods are placed at the extremes of the line where the estimation of air average temperature is required. One tripod carries the loudspeaker and the second one a microphone. The distance between the two, *d_m_*, is measured with the aid of a calibrated electronic distance meter (EDM) with a relative uncertainty of 10^−4^. The acoustic distance, *d_a_*, is estimated according to Equation (2). An auxiliary laser is placed on the microphone and is used to align it towards the loudspeaker.

**Figure 4 sensors-23-01638-f004:**
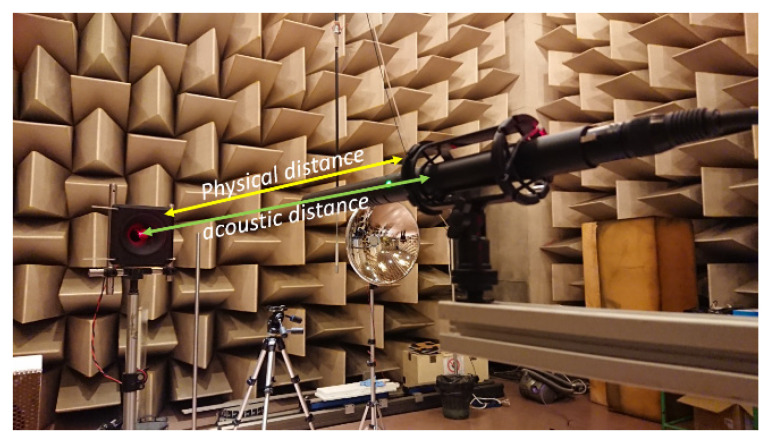
Picture of the experimental setup. In the foreground the directional microphone is visible. On the left hand side, the piezoelectric loudspeaker is visible. The red light dot in the loudspeaker is a laser pointer used to aim at the microphone when placed at large distances.

**Figure 5 sensors-23-01638-f005:**
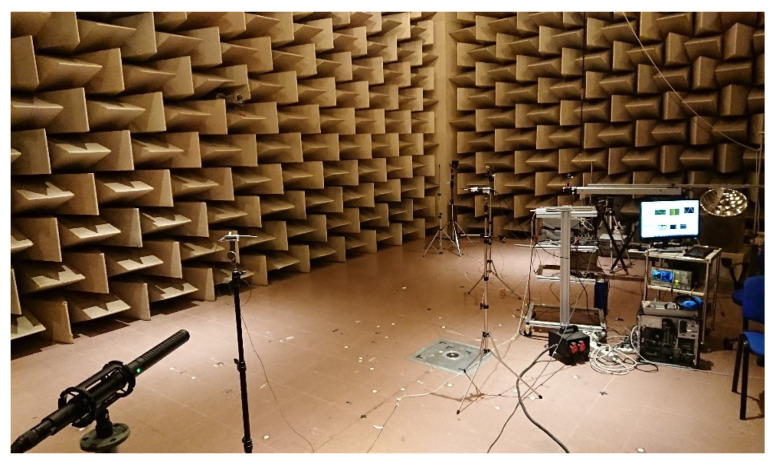
Picture of the setup for the experiment described in this chapter.

**Figure 6 sensors-23-01638-f006:**
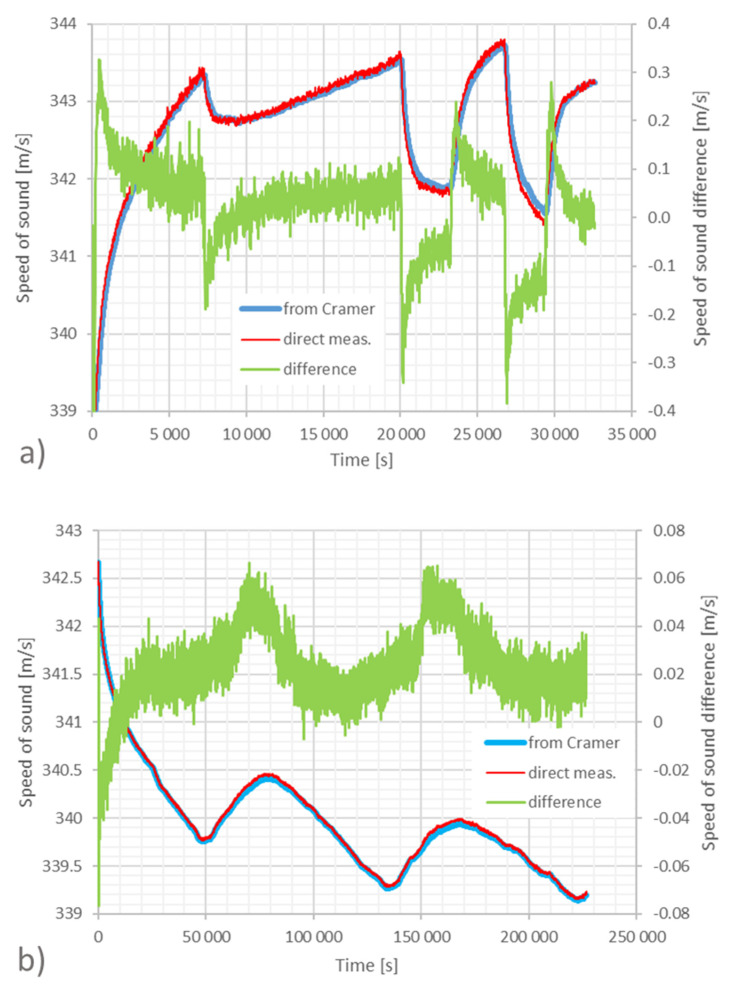
Comparison between the speed of sound measured by the AT and that calculated by the Cramer equation from the temperature obtained by classical thermometers over an 8.2 m distance in two different measurements: in (**a**) the heating plant is switched on and off to force temperature changes; in (**b**) the heating system is off and oscillations are due to the external day-night temperature changes. The blue curves are the speed of sound calculated from the average of the temperature measured by the four thermometers placed along the acoustic path. The red curves are the speed of sound directly measured by the AT. Both curves refer to the left scale in m/s. The green curves are the difference between the two referred to the right scale in m/s. The maximum errors occur when the temperature change is faster and is due to the slow reaction time of the PT100 compared to the AT.

**Figure 7 sensors-23-01638-f007:**
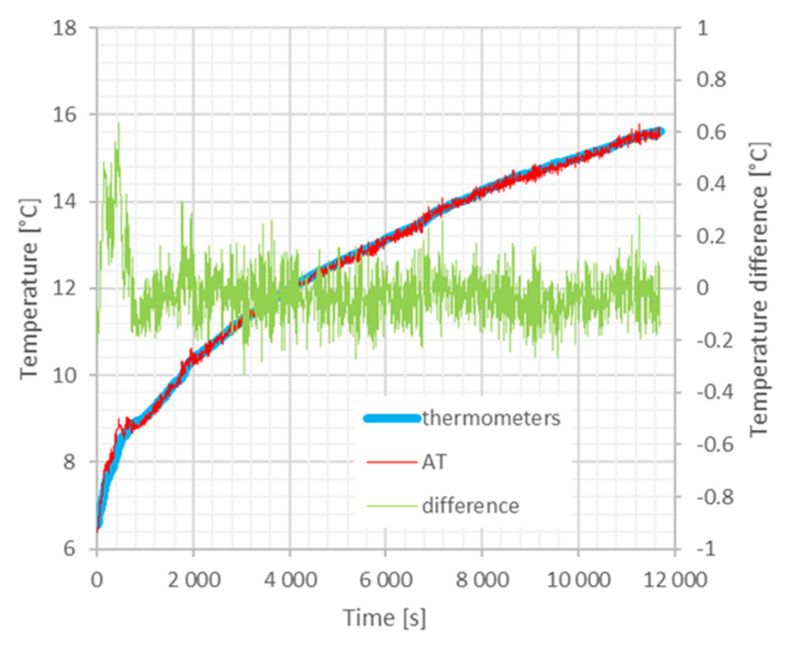
Comparison between the temperature measurement obtained by platinum thermometers and the acoustic thermometer over a 11 m distance. The blue curve is the mean of the temperature measured by the four thermometers placed along the path. The red curve is the temperature estimated from the speed of sound through the Cramer equation. Both curves refer to the left scale in °C. The green curve is the difference between the two referred to the right scale in °C. The maximum error occurs when the temperature change is faster and is due to the slow reaction time of the PT100 compared to the acoustic thermometer.

**Figure 8 sensors-23-01638-f008:**
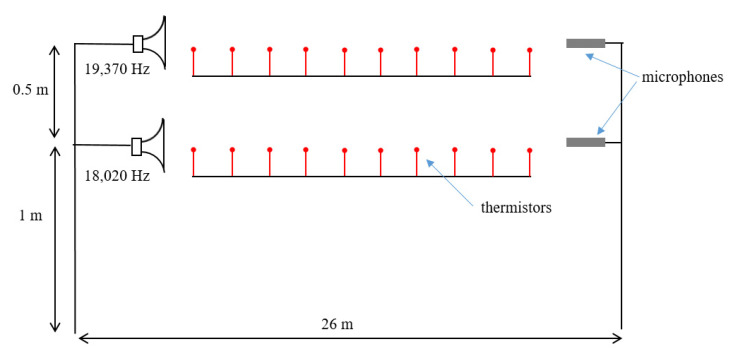
Sketch of the experimental setup for the measurement of the vertical temperature gradient.

**Figure 9 sensors-23-01638-f009:**
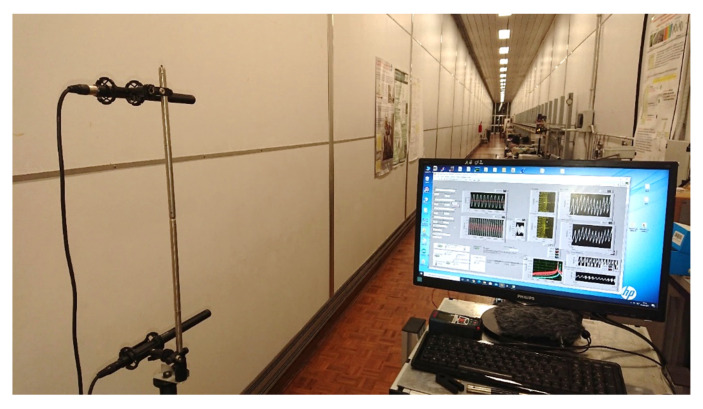
Picture of the experimental setup for the measurement of the vertical temperature gradient. In the foreground the two microphones and the screen of the LabView program used to analyze the data. The metrological gallery is 70 m long. Experiments at 60 m and at 26 m were performed successfully.

**Figure 10 sensors-23-01638-f010:**
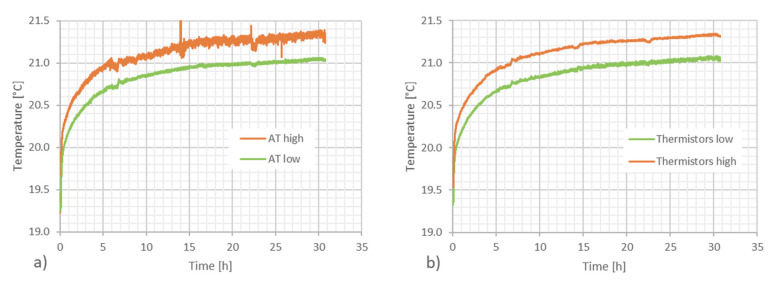
Temperature at two distances from the floor (1 m and 1.5 m) during a period of about 30 h when the stabilization circuit of the laboratory is switched off. (**a**) The temperature measured with the acoustic thermometers (AT) along a distance of 26 m. (**b**) the temperature measured in the same area with a series of 20 thermistors distributed along the path.

**Figure 11 sensors-23-01638-f011:**
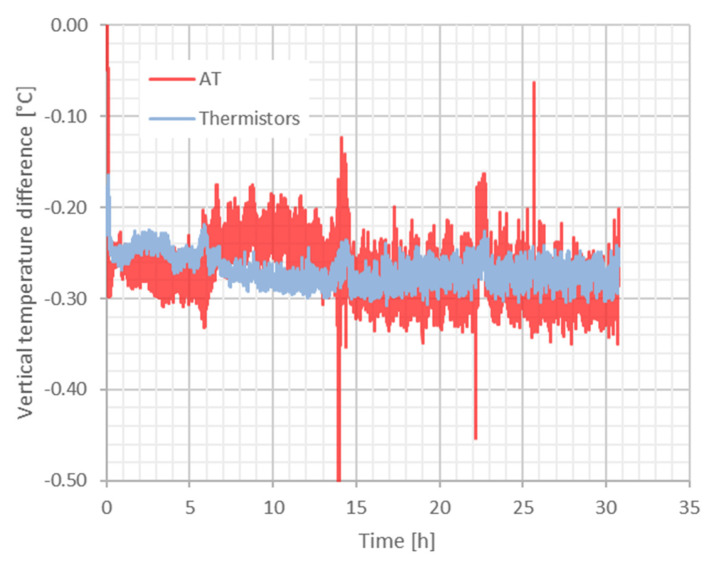
The vertical temperature difference measured by the two systems. An average gradient of about 0.3 °C/0.5 m is observed in the 30 h period. The agreement between the systems is better than 0.1 °C.

## Data Availability

Raw data used to produce the results published here are available at https://doi.org/10.5281/zenodo.7593564.

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
