# Peer review of "Improved Acoustic Thermometry for Long-Distance Temperature Measurements"

_sensors, 2023, doi:10.3390/s23031638_

Round 1

Reviewer 1 Report

The authors describe how they installed a TA measurement setup and their resolution. Overall the paper is easy to follow but its soundness is more like a report and not a scientific paper. I missed the scientific soundness, the quantification of results, the confirmation of their results (cross-check), and also the reason for the installation of the setup.

The authors should also re-check the consistency of the abbreviations. 

Here are some more precise suggestions for improving the paper:

Line 60-63: Naming the project is unusual - should be just noted in the acknowledgment

Equation 1: f is not explained

Figure 1: What is ADC?

Table 1: there is no scientific soundness - it is too objective - the authors should quantify their terms in the table

Line 159: The calibration should be more explained. Is it really important to know where it is calibrated? It is more interesting to know: How it was calibrated and which instruments were used?

Figure 3: it would be good to sketch also d_m and d_a

Figure 5a: This figure is just worthless for the reader

Figure 5b: Is the , the separator of the digit? --> then use a dot instead

Line 187: Why is the date important?

Line 191: What was the value for P and RH? Should be plotted as well

Figure 6: What info do you want to give to the reader? Is that figure really important

Figure 7: How was the thermoresistor switched on/off? (a)(b) y-axis should be the same - name the graph (a) (b) 

How was RH, P and T measured in the experiment?

Figure 9: A schematic would be better than that picture

Line 282: Why use the term "confident" --> quantify it

Figure 10 (a)+(b): the legend is not clear for me

Author Response

We thank the reviewer for his punctual observation that allowed us to improve the manuscript substantially. Please see the attachment where we tried to answer point by point.

Reviewer 2 Report

The manuscript presents an interesting work to estimate ambient temperature of air based on the acoustic transmission test where the effective acoustic and mechanical properties of air is highly temperature dependent. In general, the work has nice quality and is practically useful. The reviewer recommends this work for publication with some requests on additional discussions.

1.       Some discussion needs to state the limitations on the acoustic temperature measurement. Please add some discussion on how the environmental air flow produced by AC or other sources will impact the accuracy on the acoustic transmission test due to Doppler effect.

2.       Since the application in industrial is claimed be important, please add some discussion on how the dusts will impact the feasibility on the acoustic transmission test due to the increase of the attenuation and dispersion.

3.       The abstract was slightly confused when the reviewer read it at the beginning. If possible, please rewrite the abstract starting with a clear statement of the main objective of the work.

4.       Some figures are not in good condition. Please replace Figure 5 a with a high-resolution version. Please adjust the X-axis on Figure 5 b to the range from 10k to 20k. Please mark the difference on the right Y-axis on Figures 7 and 8. Please add Y-axis to Figures 10 and 11.

5.       In conclusion or discussion, please provide a normalized representation on the temperature accuracy (normalized by the room temperature) and max distance (normalized by the operating acoustic wavelength) for audience. 

Author Response

(The authors gave the same response as above.)

Round 2

Reviewer 1 Report

There are just style issues left:

I would highly recommend to have the same style for every graph. 

The paper looks not professional when every graph has its own format. 

Eg: There are some graphs with grid lines some with none. Also in some figures the axis label is written in upper letters in some not etc.

In the last figure, the legend has still the autocorrection line.

Author Response

We thank the referee for the comments, indeed now the paper looks more professional. Here are the answers to the comments.

I would highly recommend to have the same style for every graph. 

> We have modified all pictures with graphs, namely 6, 7, 10 and 11, trying to maximise uniformity in style

The paper looks not professional when every graph has its own format. 

Eg: There are some graphs with grid lines some with none. Also in some figures the axis label is written in upper letters in some not etc.

> Labels have been corrected

In the last figure, the legend has still the autocorrection line.

> Corrected